# Benefits of Core–Shell Particles over Single-Metal Coatings: Mechanical and Chemical Exposure and Antimicrobial Efficacy

**DOI:** 10.3390/polym16152209

**Published:** 2024-08-02

**Authors:** Sabine Poelzl, Stefan Augl, Alexander Michael Schwan, Simon Chwatal, Jürgen Markus Lackner, Clemens Kittinger

**Affiliations:** 1Diagnostic and Research Institute of Hygiene, Microbiology and Environmental Medicine, Medical University of Graz, Neue Stiftingtalstraße 2A, 8010 Graz, Austria; sabine.poelzl@medunigraz.at; 2Department of Materials Technology, University of Applied Sciences Upper Austria, Stelzhamerstraße 23, 4600 Wels, Austria; stefan.augl@fh-wels.at; 3MATERIALS—Institut für Sensorik, Photonik und Fertigungstechnologien, Joanneum Research Forschungsgesellschaft mbH, Leobner Strasse 94a, 8712 Niklasdorf, Austriasimon.chwatal@joanneum.at (S.C.); juergen.lackner@joanneum.at (J.M.L.)

**Keywords:** antimicrobial coating, copper, zinc, core–shell, durability test, *S. aureus*, bacteriophage ϕ6

## Abstract

One of the greatest challenges worldwide is containing the spread of problematic microorganisms. A promising approach is the use of antimicrobial coatings (AMCs). The antimicrobial potential of certain metals, including copper and zinc, has already been verified. In this study, polyethylene terephthalate and aluminum (PET-Al) foils were coated with copper, zinc, and a combination of these two metals, known as core–shell particles, respectively. The resistance of the three different types of coatings to mechanical and chemical exposure was evaluated in various ways. Further, the bacteria *Staphylococcus aureus* and the bacteriophage ϕ6 were used to assess the antimicrobial efficacy of the coatings. The best efficacy was achieved with the pure copper coating, which was not convincing in the abrasion tests. The result was a considerable loss of copper particles on the surfaces and reduced effectiveness against the microorganisms. The core–shell particles demonstrated better adhesion to the surfaces after abrasion tests and against most chemical agents. In addition, the antimicrobial efficiency remained more stable after the washability treatment. Thus, the core–shell particles had several benefits over the pure copper and zinc coatings. In addition, the best core–shell loading for durability and efficacy was determined in this study.

## 1. Introduction

Infections caused by pathogens are one of the main causes of chronic infections and mortality. According to the World Health Organization (WHO), the deadliest communicable disease is the infection of the lower respiratory tract, which is ranked as the fourth leading cause of death worldwide [1]. Although medical standards including hygiene measures and medical treatments are already very high nowadays, industrialized countries are still facing major issues. Viral outbreaks such as the SARS-CoV-2 pandemic and multi-resistant bacterial strains prove humankind’s inferiority to the natural evolution of microorganisms. At the beginning of the pandemic in 2020, little was known about the coronavirus; thus, suitable treatment options were scarce, and preventative measures through vaccinations needed time to be developed. The problem of antimicrobial resistance (AMR) has been caused by the disproportionate and, in most cases, inappropriate use of antibiotics in infection treatment or in agriculture and livestock [2], respectively. Subsequently, multi-drug-resistant strains have emerged. The above issues are just two examples of why additional support and further medical developments are needed to contain the spread of coronaviruses and other problematic microorganisms. One promising approach is the use of antimicrobial coatings (AMCs) since they exhibit rapid neutralization of microorganisms without further effort. Especially in vulnerable institutions like healthcare systems containing patients with serious health issues and impaired immune systems, AMCs could protect against additional infections transmitted through infected and frequently touched surfaces. In addition, implant-associated infections are also a huge medical issue [3]. The application of AMCs has been established in many studies by animal models as well as clinical trials and showed protection against the development of infection in many cases. However, rising concerns regarding the long-term effectiveness of AMCs have prevented this application so far [4]. In addition, because of the rapidly increasing infections with antibiotic-resistant bacteria, the application of AMCs is viewed critically. There are legitimate concerns that bacteria could become resistant to such coatings because of the potential selection pressure by different processes (e.g., de novo mutation, horizontal gene transfer, and species sorting of inherently resistant bacteria) [5]. The concern of prompting and spreading of AMR is especially present in healthcare facilities. However, most studies are not accurately evaluated for these environments since experiments were conducted in solution or high-humidity conditions. Therefore, a clear consensus on this issue is not available yet. Pietsch et al. [5] concluded that the benefits of AMCs in healthcare settings might outweigh the risks of generating AMR.

Another area for applying antimicrobial coatings is, rather unexpectedly, the packaging sector. Preservation methods for packed food are no longer limited to pasteurization, sterilization, cooling, freezing, high-pressure or cold plasma processing, irradiation, and other processes [6]. Moreover, antimicrobial food packaging is used increasingly to inhibit the growth of microorganisms, especially to reduce pathogens and control spoilage microorganisms [7] without altering the food itself [8]. This additional method of preserving food seems particularly beneficial, considering that some packaging materials can provide a thriving environment for bacteria. For instance, Schmid et al. [9] generated a laboratory model to evaluate the growth and survival of bacteria within fiber-based packaging material. Based on this model, certain species could grow up to 10.8 log_10_ CFU/mL within six days. Therefore, some studies have already investigated antimicrobial food packaging as summarized by Y. Fu and E. Dudley [10]. Metal and metal oxide nanoparticles have especially gained great interest recently. The study of Azlin-Hasim et al. [11] showed that silver nanoparticles (AgNPs) could extend the shelf life of packed chicken from 6.5 to 9 days at 4 °C. Zinc oxide nanoparticles (ZnONPs) could also increase the shelf life from up to 12 days for okra at 25 °C [12] and can lead to an approximately 4 log_10_ reduction in classic foodborne pathogens in nutrient broth after 24 h at 37 °C [13].

As mentioned, various metals show antimicrobial efficiencies and can be used for coatings. Copper is the most commonly used metal as an AMC, with well-established efficacy and broad applications [5]. In addition, copper was the first metal to be approved as an AMC by the Environmental Protection Agency (EPA) in 2008 [14]. Nowadays, zinc is also reputed to have antimicrobial properties, especially in the form of complexes or zinc oxide nanoparticles [15]. The mechanism behind the antimicrobial effect is not fully understood yet. However, the redox properties of the metals and the tendency to transit between the oxidation states (e.g., cuprous [Cu^1+^] and cupric [Cu^2+^] states) are attributed to the antimicrobial effect. The main killing mechanism is the release of reactive oxygen species (ROS) and not the metal ions or a nanoparticle effect. ROS include the superoxide radical (O_2_•–), the hydroxyl radical (·OH), hydrogen peroxide (H_2_O_2_), and the singlet oxygen (O_2_). Copper oxide can generate all four types of ROS, while zinc oxide can only lead to H_2_O_2_ and ·OH. The oxidative stress induced by ROS leads to changes in cell membrane permeability, to an attack of proteins and, thus, their depression of activity, and to increased gene expression of oxidative proteins, which can ultimately trigger apoptosis [16]. Although metal ions can also contribute to the reduction in enzyme activities, changes in cell structure, and affect physiological processes, they are not considered very important for antimicrobial activity [17].

In recent years, atmospheric pressure plasma has become a reliable and popular coating method [18,19]. This innovative approach involves the introduction of precursors and/or powders into the plasma, melting them, and then depositing them with a diverse range of parameters. By adjusting plasma parameters such as gas flow, current intensity, and coating speed, the loading density on substrates can be modified to achieve the desired properties. Notably, this method is suitable for coating sensitive materials such as thin plastic films or bio-based materials at atmospheric pressure. The plasma temperature can be precisely adjusted to enable this flexibility.

Due to the high velocity of particle impact on substrate surfaces, large particle-laden spots are formed. The size of these spots depends on the melting temperature of the particles. For instance, zinc particles form larger spots than copper particles. The larger reactive surface that results from these spots boosts the antimicrobial effect of coatings.

In our study, we investigated the antimicrobial efficiency of copper and zinc alone, and a combination of both metals, since synergistic effects of different metals have also been reported before. For instance, Garza-Cervantes et al. [20] demonstrated that combining silver with other metals including copper and zinc increases the antimicrobial effect up to 8-fold while the concentration is low/non-toxic for eukaryotic cells. This study focuses on core–shell particles consisting of a zinc core and a thin copper shell to investigate the antimicrobial efficacy of this combination. Core–shell nanoparticles are already used extensively in biomedical applications, including bioimaging, drug delivery, gene delivery, and sensors [21], and some combinations (e.g., Au/Ag; ZnO/SiO_2_) have also been investigated for their antimicrobial potential [22,23]. However, a copper–zinc combination has not yet been tested for durability by different abrasion tests and antimicrobial efficiency. Therefore, this study aimed to determine the correct core–shell particle loading percentage for optimal durability and efficacy. Furthermore, the durability and resistance of all generated coatings against mechanical and/or chemical exposure were evaluated visually and also against selected microorganisms to guarantee a long-lasting product with antimicrobial properties.

## 2. Materials and Methods

### 2.1. Coating and Samples

As substrate for the coatings consisting of a compound system of polyethylene terephthalate (PET) and Aluminum (PET-Al) was used. PET served as a coating site. The foils were coated with commercially available copper (Cu, average diameter = 25 µm) and zinc (Zn, average diameter = 30 µm) particles as well as a combination of both metals known as core–shell (CS, average diameter = 35 µm) particles, which had a Zn core and a thin Cu shell (0.2–0.4 µm). For Cu and Zn, only one particle loading of each was used for further experiments. The CS specimens were generated with different concentrations of CS particles to identify the correct particle loading percentage for good durability and antimicrobial effect (Table 1).

Eckart GmbH (Hartenstein, Germany) provided the test powders for the coating and the coating itself was conducted through Atmospheric Pressure Plasma Deposition (APPD) using an INOCON InoCoat3 Plasma Jet (INOCON Technologie GmbH, Attnang-Puchheim, Austria). In this coating technology, the argon plasma is ignited by means of an electric arc with a voltage of approximately 22 V between a tungsten cathode and a copper anode. The output is approximately 2 to 11 kW, depending on the current used. The powder required for coating is fed into the plasma nozzle via a powder brush conveyor and can be fed into the plasma directly after the plasma ignition point (Figure 1). By varying the gas flow and current intensity, the powder output can be controlled and the degree of melting of the particles can be adjusted.

### 2.2. Particle Composition and Coating

Based on the EDX analyses of the layers, oxide layers can be excluded with a high degree of probability when using pure copper or zinc powder. The element distribution on the substrate shows pure copper and zinc areas. Only a small amount of oxygen was detected in these areas. The concentration indicates pure, non-oxidized metal (Figure 2). The “dissolution” of the particles is more likely to be a flaking of areas of the coating, which was caused mechanically. When using the core–shell particles, the copper shell is present as an oxide. This was again confirmed by SEM and EDX images.

The concept of the CS particles is intended to combine the silvery, shiny appearance of Zn coatings with the antimicrobial functionality of Cu coatings. Four variants with different particle loadings (Table 1) were provided by Eckart GmbH. Surface analysis was carried out with GIMP 2.10.24 to define Cu, Zn, and uncoated areas by means of color matching using grey scales (Figure 3). On average, this resulted in an almost 50% loading of the surface with particles, of which 30% were Cu particles (Cu/Zn ratio: 0.61).

In the next step, sections of the powders were prepared and an attempt was made to determine the thickness of the copper shell. Figure 4 shows a thickness of about 100 nm and the chemical composition of the shell. The total size of the particle was approximately 35 µm. This means that the zinc concentration is significantly higher than the copper concentration.

A SEM image with material contrast shows that the burst copper shell remains on the zinc particle after the coating process (Figure 5). The chemical analysis shows that the coating is very thin, as a lot of zinc is also detected due to the depth of excitation.
Figure 5SEM image + EDX analysis of the burst copper shell of a CS particle after coating. Scanning electron microscopy (SEM) was used to improve the accuracy in determining the particle loading. The material-dependent signal intensity of the back-scattered electrons (BSEs) allows an image with clear contrast by using different grey levels of film and coating. The proportion of coating per image section can be calculated by analyzing the area (Figure 6). It should be noted that the volume is not taken into account here. The percentage distribution of the particles on the surface is shown in Table 1.
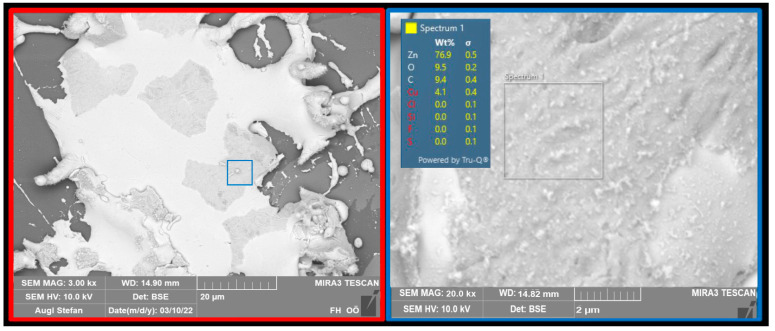

Figure 6An example for measuring the particle load for CS2 in SEM (BSE mode with 7.5× magnification) and further depiction of the particle distribution of pure copper (Cu1) and pure zinc (Zn1) coatings.
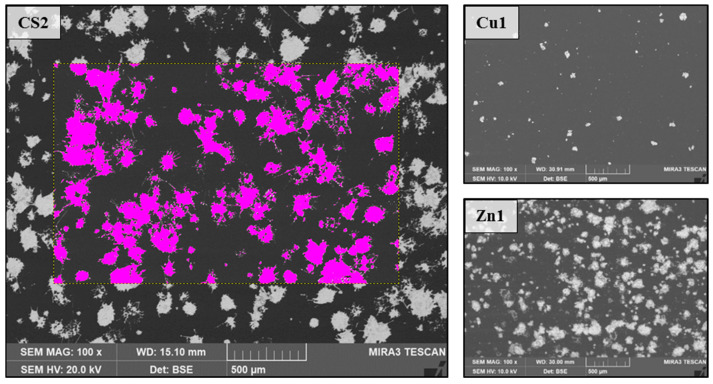



### 2.3. Durability Analyses of the Coating

#### 2.3.1. Mechanical Exposure—Crockmeter Test

The abrasion durability test was conducted using an electronic crockmeter TF411 (TESTEX, Dongguan, China) based on ISO 105-X12:2016 [24]. This standard is intended for testing color changes in textiles under frictional stress. Therefore, the evaluation method of the standard had to be adapted accordingly for testing coated plastic films. An examination by light microscopy was not possible since the shiny silver zinc particles offer too little contrast on the shiny metallic film (the aluminum of the composite shines through the PET in the light microscope) to make a distinction and thus assess the abrasion. Therefore, the abrasion was measured in the same way as particle loading using the BSE mode of SEM with 7.5× magnification. The abrasion was performed by a 16 mm rubbing head running 1000 cycles with 9 N over the surface of 150 × 50 × 0.365 mm samples.

#### 2.3.2. Chemical Exposure—Cleaning Agents

The CS coatings were tested on their resistance against chemicals. The chemical resistance tests were carried out following the EN 12720:2009+A1:2013 Furniture—Assessment of surface resistance to cold liquids [25]. Therefore, a soaked cotton pad with test liquids was placed on the samples and everything was covered with a container. After one hour, the cotton pad and the container were removed, leaving the samples in a certain test climate for 24 h. After one day of incubation, the samples were cleaned and allowed to dry before being analyzed by a stereo microscope. Common household cleaners were used as test liquids (Table 2). The selection of test liquids covered a wide range of cleaning agents with pH values ranging from 2.4 (Frosch-Essig Reiniger) to 11.4 (CILLIT BANG—FETTLÖSER Kraftreiniger).

After the chemical exposure treatment, the surfaces of the samples were assessed by a stereo microscope and categorized numerically (Table 3).

#### 2.3.3. Mechanical and Chemical Exposure—Washability Test

A washability test of the Cu1, Zn1, and CS3 samples was performed to test the mechanical and chemical exposure treatments using the Scrub Abrasion and Washability Tester AB6000 from TQC sheen (Industrial Physics LLC, Boston, MA, USA). For the test, weighted brushes were moved over the samples (25 × 50 × 0.365 mm) at a certain speed or frequency while washing solutions (e.g., water, cleaning agents) were dripped onto the tracks at a rate of 1 mL/min. A wash cycle corresponds to a forward and backward movement of the brush over the entire length of the lane. The standard settings are based on ASTM D2486 [26], such as the weight of the nylon brush (454 g), frequency (37 cycles/minute), and length of the lane (300 mm). Differences were present in the total number of cycles (up to 10,000 cycles ≙ 6000 m) and the applied washing solutions. Either distilled water (ddH_2_O) or commercially available acetic acid cleaning agent (Frosch-Essig Reiniger, abbreviated throughout the text as AACA) were used as washing solutions. An amount of 6 mL of AACA with pH~2.4 was diluted in 500 mL of ddH_2_O for the application. The samples were analyzed optically under a stereo microscope with energy-dispersive X-ray spectroscopy (SEM-EDX). The number of particles, the particles themselves (shape and composition, e.g., oxidation), and the substrate surface were analyzed for changes after the washability treatment.

### 2.4. Antimicrobial Surface Tests

All samples with the dimensions of 50 × 50 × 0.365 mm were produced aseptically and individually packaged in a plastic film to ensure sterility during transportation to the microbiological laboratory. If a pre-treatment (washability test) was performed, the samples with the dimensions of 25 × 50 × 0.365 mm were re-sterilized with 70% ethanol prior to antimicrobial testing.

After the antibacterial and antiviral tests were carried out, two different assays were used to analyze the antimicrobial efficiency.

#### 2.4.1. Testing of Antibacterial Activity

The test principle is based on ISO 22196:2011 [27] and the following procedure was used: The bacterial strain *Staphylococcus aureus* (*S. aureus*) DSM 346 (Leibniz Institute DSMZ—German Collection of Microorganisms and Cell Cultures GmbH, Braunschweig, Germany) served as testing organism. The cultivation was carried out overnight on Columbia Blood agar plates (Becton Dickinson Austria GmbH, Vienna, Austria) at 36 °C ± 2 °C. Cell material was inoculated in a 1/500 tryptic soy broth (TSB) (Oxoid Ltd., Basingstoke, United Kingdom diluted in distilled water. A VITEK^®^ DensiCHEK instrument (Biomerièux, Vienna, Austria) was used to obtain a bacterial solution with 1 × 10^8^ colony-forming units (CFUs)/mL. An amount of 150 µL or 100 µL (washability test) of bacterial suspension with an expected bacterial concentration of 2.5–10 × 10^5^ cells/mL was pipetted on the sterile surfaces of the samples. The suspension was covered and distributed under a sterilized 40 × 40 mm or 20 × 40 mm (washability test) polyethylene film (VWR International, Vienna, Austria). After that, the inoculated test specimens were incubated in a wet chamber at 36 °C ± 2 °C with a relative humidity (RH) of approximately 96% for 10 min (min), 30 min, and/or 1 h (h). As a control, the bacterial suspension was harvested immediately after pipetting below the foil (0 h) to ensure initial concentration from each specimen and to validate loss through manipulation. After the exposure treatment, the samples were rinsed off the surface by four washing steps with 10 mL of neutralizer (SCDLP medium) containing TSB (Oxoid Ltd., Basingstoke, United Kingdom), lecithin (Carl Roth GmbH + Co Kg, Karlsruhe, Germany), and Tween^®^80 (Amresco Inc., Solon, OH, USA) in order to rescue surviving bacteria. The samples were then shaken in the neutralization medium for 3 min at 200 rpm on a Battery Shaker KM 2 Akku (Edmund Bühler GmbH, Bodelshausen, Germany). An amount of 500 µL of appropriate dilutions in phosphate-buffered solution (PBS) (Carl Roth GmbH + Co Kg, Karlsruhe, Germany) was plated on tryptic soy agar (TSA, VWR International Ltd., Vienna, Austria) plates in duplicates. CFUs were counted after incubation for 24 h at 36 °C ± 2 °C.

#### 2.4.2. Testing of Antiviral Activity

The surfaces were tested for their antiviral activity with the bacteriophage Phi6 (ϕ6) DSM 21518 (Leibniz Institute DSMZ—German Collection of Microorganisms and Cell Cultures GmbH, Braunschweig, Germany). The bacteriophage propagation was conducted according to the manufacturer’s specifications. In brief, the host strain *Pseudomonas syringae* (*P. syringae*) DSM 21482 (Leibniz Institute DSMZ—German Collection of Microorganisms and Cell Cultures GmbH, Braunschweig, Germany) was cultivated overnight in lysogenic broth (LB, Carl Roth GmbH + Co Kg, Karlsruhe, Germany) containing CaCl_2_ (Merck KGaA, Darmstadt, Germany) at 25 °C and 110 rpm. A 1:50 dilution of the overnight culture was again grown in 10 mL of fresh LB media to OD_600_ 0.2 determined by a UV/VIS spectrophotometer (VWR International GmbH, Vienna, Austria). The culture was infected with an approximate multiplicity of infection (MOI) of 0.1 and then incubated for 4 h at 25 °C and 50 rpm. The suspension was stored at 4 °C overnight. On the next day, the phage lysate was centrifuged (10,000× *g*, 20 min, and 4 °C) and then filtered through a sterile syringe 0.2 µm filter (VWR International GmbH, Vienna, Austria). The phage suspension was stored at 4 °C and the titer was determined by plaque assay.

The experiments were carried out according to ISO 21702:2019 [28], using the bacteriophage plaque assay according to ISO 18071:2016 [29]. The sterile surfaces were inoculated with 150 µL or 100 µL (washability test) of viral suspension with an expected viral titer of 1–4 × 10^7^ plaque-forming units (PFUs)/mL and were then covered by a sterilized 40 × 40 mm or 20 × 40 mm (washability test) polyethylene film. Immediately after inoculation (0 h) as well as after 10 min, 30 min, and/or 1 h of exposure at 36 °C ± 2 °C in a wet chamber (~96% RH), remaining infectious viral particles were recovered by using 10 mL of SCDLP medium. After washing four times, dilutions in peptone saline solution (Carl Roth GmbH + Co Kg, Karlsruhe, Germany) were prepared. Amounts of 0.1 mL of the appropriate host and 1 mL of the viral dilution were added to 2 mL of liquid Top Agar containing peptone, saline, yeast extract, and agar (Carl Roth GmbH + Co Kg, Karlsruhe, Germany) with the addition of CaCl_2_ (Merck KGaA, Darmstadt, Germany). After gently mixing, the solution was poured over LB agar (Carl Roth GmbH + Co Kg, Karlsruhe, Germany) containing CaCl_2_. Each dilution was determined in duplicates. PFUs were counted after incubation for 24 h at 25 °C ± 2 °C.

When no colonies or plaques were countable on the plates, the limit of detection was set as 10 CFU/PFU, since 10 mL of the neutralization medium was used. For each incubation time, triplicates (n = 3) were used to calculate the mean and standard deviation for the antibacterial/antiviral activity of the samples. The test validity was calculated through 0 h triplicates with the following formula: (LOGmax − LOGmin)/LOGmean ≤ 0.2. A value below 0.2 indicated a valid test result.

Since the evaluation differs between the used ISOs, an equal calculation of the results was chosen to make it comparable. Therefore, the calculation is based on ISO 18071:2016 [29].

### 2.5. Graphics and Statistical Analysis

All statistical analyses and graphical depictions were performed with GraphPad Prism 9.

## 3. Results

### 3.1. Durability Analyses of the Coatings

#### 3.1.1. Mechanical Exposure—Crockmeter Test

The crockmeter test was used to examine all samples for mechanical resistance against a 16 mm rubbing head, which ran over the surface of each specimen for 1000 cycles with a vertical pressure of 9 N. Since the aluminum of the foil and Zn particles are very similar in gloss and color, an assessment of the abrasion in the stereo microscope was difficult. Therefore, the abrasion was determined by SEM (Figure 7).

By comparing the particle loading percentage before the crockmeter was used, a relatively high abrasion on Cu1 was detectable after the treatment. Even though the Cu coating already had a very low loading of 1.08%, the proportion of Cu particles was reduced to only one quarter after mechanical treatment. Moreover, 11.35% of Zn particles were lost due to mechanical exposure on Zn1 samples. In comparison, the CS samples showed better resistance within the crockmeter test. For instance, specimens with a CS particle loading of 10.13% (CS3) showed only a 1.66% reduction in coating particles on the observed surfaces. Even CS1 with the highest core–shell loading of all tested samples lost only 2.55% of particles through the crockmeter test. Therefore, a weaker adhesion of the pure Zn coating of Zn1 compared to the CS coating with approximately the same particle load (CS1) was detectable. The particles of CS4 even increased on the surface from 5.02% to 7.20%, which was caused by flattening the particles by applying the crockmeter (Figure 8).

#### 3.1.2. Chemical Exposure—Cleaning Agents

The chemical exposure treatment to common household cleaning agents was performed with all four different particle loadings of the CS coatings. The following cleaning agents were used: Frosch-Essig Reiniger (Werner & Mertz GmbH, Mainz, Germany), Cif-Bad & Dusche Salle de Bain (Unilever Austria GmbH, Vienna, Austria), Frosch-Glas-Reiniger Spiritus (Werner & Mertz GmbH, Mainz, Germany), Frosch-Aktiv-Soda-Reiniger (Werner & Mertz GmbH, Mainz, Germany), CILLIT BANG-FETTLÖSER Kraftreiniger (Reckitt Austria GmbH, Vienna, Austria). The durability against acidic chemicals was insufficient, while alkaline cleaning agents showed no high abrasion of the CS coatings. An influence of the particle loading could not be determined in these tests (Table 4).

A comparison of good and weak chemical resistance is illustrated in Figure 9 using CS4 as an example. While the vinegar-based cleaner (Frosch—Essig Reiniger) completely removed the particles of CS4, all particles are still intact in the case of the power cleaner (CILLIT BANG—Kraftreiniger).

#### 3.1.3. Mechanical and Chemical Exposure—Washability Test

During the washability process, Cu1, Zn1, and CS3 surfaces underwent exposure to either ddH_2_O or a diluted acetic acid cleaning agent (AACA, Frosch-Essig Reiniger with pH 2.4 was used, see Table 2) while enduring up to 10,000 mechanical abrasion cycles. The progression of abrasion was analyzed using a microscope after a set number of cycles (Figure 10). In an acidic environment with AACA, all coatings experienced severe abrasion after 1000 cycles. The Cu and CS coatings remained relatively unscathed, whereas the pure Zn particles of Zn1 exhibited a noticeable change in appearance and distribution on the surface. Conversely, treatment with ddH_2_O resulted in little to no abrasion. After 2500 cycles with ddH_2_O, all visible particles at 4× magnification displayed strong adhesion and resistance to abrasion, with no signs of wear detected. An optical change occurred only after the highest number of cycles (10,000 cycles), particularly for the particles of Zn1 and CS3.

In order to conduct a thorough analysis, we utilized a scanning electron microscope (SEM) to capture images and then employed energy-dispersive X-ray spectroscopy (EDX) to analyze the images (Figure 11, Figure 12 and Figure 13). The primary focus was on the examination of untreated samples of Cu1, Zn1, and CS3 as well as samples that underwent 10,000 cycles in a washability tester using water or an acetic acid cleaning agent.

Similar to the microscope examination, a notable difference was observed after AACA treatment. The Cu1 sample demonstrated the highest degree of stability. However, no coating particles were detected on the Zn1 and CS3 samples. The impact of the plasma on the powder used was also clearly evident. It correlated with the different melting points as the Zn1 and CS3 powders have a considerably lower melting point (approximately 419 °C) than the Cu1 powder (1083 °C).

Consequently, the deformation of the Zn1 and CS3 powders, which is clearly visible in both SEM and EDX images, was expected. This deformation resulted in significantly larger clusters, with a larger reactive surface area than the Cu1 samples. However, the clusters were not well anchored to the substrate, which explains why the coating did not adhere well under the harsh test conditions (AACA). Further, because of the low pH of the AACA solution (< 3), the zinc and CS particles must have been oxidized and dissolved in the acidic environment, which is in accordance with our results of the chemical exposure test (Table 4).

### 3.2. Antimicrobial Efficiency

#### 3.2.1. Comparison of Three Different Coatings

The Cu, Zn, and CS coatings were tested for their antimicrobial efficiency. An inoculum with a certain concentration (see applied load, Figure 14) of *S. aureus* and bacteriophage ϕ6 was incubated on the PET-Al foils with and without (reference) a coating for up to one hour at 37 °C and an RH of ~96%. After incubation, a neutralizer medium was used for harvesting and the infectivity of the microorganisms was verified by a plating technique or plaque assay. In the case of Cu1, a complete efficiency against *S. aureus* and ϕ6 was already detectable after 30 min. Furthermore, the short incubation of 10 min on the Cu coating already achieved a good reduction of 1.7 log_10_ colony-forming units (CFUs) and of 4.3 log_10_ plaque-forming units (PFUs) compared to the initial concentration (0 h). The Zn and CS coatings required twice as long as Cu1 for an almost complete reduction in *S. aureus*. In the case of Zn1, one CFU and, for CS3, three CFUs were detectable after one hour. The antiviral effect on CS3 was similar to the antibacterial effect. After one hour of incubation on this coating, no plaques were detected. The Zn1 results between *S. aureus* and ϕ6 were not comparable. While a complete bacterial efficiency was detectable after one hour of incubation, 9.25 × 10^3^ PFU were still detectable for ϕ6. The uncoated reference showed no notable reduction over the entire incubation period against *S. aureus* or ϕ6 (Figure 14).

#### 3.2.2. Comparison of the CS Coatings

All CS coatings with different particle loadings were also analyzed for their antimicrobial efficiency. The assays were performed against *S. aureus* and bacteriophage ϕ6 equally to conduct a comparison of the three different coatings above. This comparison of the different CS samples was carried out to demonstrate the influence of the particle loading on the antimicrobial effect and to achieve maximum effectiveness with minimum use of material. The results of the antibacterial assay showed no high differences between the CS samples. For 10 min, an infectious bacterial amount of 8.70 × 10^3^ to 1.76 × 10^4^ CFU was determined. The 30 min incubation time point showed a <1 log_10_ difference between the highest (CS2) and lowest (CS1) result. For 1 h, only a few CFUs were countable on the plates (two for CS1, ten for CS2, three for CS3, and thirteen for CS4), with CS4 having the most detectable CFUs of *S. aureus*. However, the output of the plaque assay showed differences between the samples. After 10 min, a difference of 1.2 log_10_ was achieved between CS1 and CS4. The highest difference could be detected after 30 min of incubation on the samples. CS1, CS2, and CS3 achieved an average load of 1.75 × 10^2^ PFU from still-infectious viral particles, while the effect on CS4 was strongly minimized to 4.04 × 10^4^ PFU. After 1 h, CS4 again showed the highest amount of still-infectious bacteriophages with 2.33 × 10^1^ PFU (Figure 15). Therefore, the test series against the various CS specimens suggests that the loading of approximately 5% CS particles (=CS4) is insufficient for optimal antimicrobial efficiency. From a cost–benefit point of view, a particle loading of approximately 10% as in the case of CS3 proved to be the best variant. Thus, only CS3 was compared to Cu1 and Zn1 for the following washability tests.

#### 3.2.3. Washability Surfaces

After the mechanical and chemical exposure of the surfaces within the washability test, the coatings were again tested for their antimicrobial efficiencies. With AACA, only 1000 washing cycles were executed in order to have an AACA setup that does not remove the zinc and CS particles completely. Since the surfaces of the specimens had to be reduced in width by half for the abrasion test, only 100 µL of the bacterial or viral suspension was applied to the samples. Therefore, untreated 25 × 50 × 0.365 mm samples of all three different coatings were again tested twice in the same run with the exposed surfaces. Thus, the 1 h incubation period per specimen was tested six times against *S. aureus* and ϕ6. For Cu1, a decreased antimicrobial effect after the washability test was observed compared to the untreated samples. The untreated samples showed a good bacterial reduction of 3.3 log_10_ and a complete viral reduction of 5.2 log_10_ compared to the initial concentration (0 h). Even though the Cu coating showed no or only less abrasion in microscopic analysis after the washability treatment (Figure 10 and Figure 11), the antimicrobial efficacy was strongly minimized. Regardless of the type of treatment (number of cycles or washing solution), the reduction in *S. aureus* and ϕ6 was less than 1 log_10_ compared to the results of 0 h (Figure 16).

No differences in efficiency between the untreated and treated surfaces against *S. aureus* could be determined for the Zn-coated specimens. All tested samples achieved a complete reduction in *S. aureus*. The exceptions of one tested replicate in each of the untreated and the water-treated (10,000 cycles) samples with one or two detectable colonies are negligible. However, the results were different for ϕ6. The antiviral effect against this bacteriophage was improved after the washability test. For the untreated surfaces, a reduction of only 3.3 log_10_ was calculated after 1 h compared to the initial concentration (0 h). Instead, the treatment with AACA and 1000 cycles achieved the best reduction of 5.3 log_10_ compared to 0 h. The treatments with ddH_2_O and 2500 or 10,000 cycles achieved a reduction between the other two results (reduction of 4.4 log_10_ and 4.8 log_10_) (Figure 17). It is conceivable that the limited efficiency of the untreated samples against the bacteriophage was compensated by the better accessibility or distribution of the particles after the washability treatment. This assumption is confirmed by the microscopic analysis (Figure 10 and Figure 12). The significant external modification during the treatments may explain the improved viral effect compared to the untreated Zn1 specimens.

Treatment with AACA showed reduced antibacterial and antiviral activity of the CS coating from CS3. Only a reduced decrease in *S. aureus* (reduction of 2.1 log_10_) and ϕ6 (reduction of 1.4 log_10_) was calculated after 1 h on the AACA-treated samples. The effect was also minimized when the abrasion was performed with ddH_2_O, but not as strongly as AACA. The number of cycles also had a major impact here. Over 1 log_10_ difference was calculated between 2500 and 10,000 cycles, with a better antimicrobial effect achieved with fewer cycles. The antimicrobial efficiency at 2500 cycles was nearly as good as on the untreated surfaces. All results (untreated and 2500 cycles with ddH_2_O) showed an approximately 4 log_10_ reduction against *S. aureus* and ϕ6 (Figure 18). These gradations of efficacy were also reflected through the microscopic examination of the treated samples (Figure 10).

## 4. Discussion

As shown in most studies, pure copper has good antimicrobial activity. Grass et al. [14] summarize that at least 7 to 8 logs per hour of different species can be killed and no viable microorganisms were generally recovered from copper surfaces after prolonged incubation. Our study confirmed these data by rapidly reducing over 4 log_10_ from *S. aureus* within 30 min (Figure 14). The antiviral effect against bacteriophage ϕ6 of Cu1 was also very similar. This bacteriophage was chosen for testing because it can be used as a model organism for several human viruses including influenza and SARS-CoV-2 [30,31]. Although the resistance of the copper coating against mechanical and chemical exposure was sufficient visually (Figure 10 and Figure 11), the antimicrobial efficiency was surprisingly reduced after the washability treatment (Figure 16). Since the Cu1 samples had only a 1.08% particle loading on the surfaces, some particles could have been washed off unnoticed, leading to a clear decrease in efficacy. This theory could also be confirmed by the results of the crockmeter test, in which the proportion of Cu particles was greatly reduced (Figure 8). The effect is possibly related to a weak interfacial adhesion strength between the copper particles and the used substrate. It appears that the copper powders did not completely melt due to their higher melting temperature, resulting in less-than-optimal adhesion to the substrate. Unfortunately, increasing the plasma temperatures to melt the copper particles would have risked damaging the substrate. On the other hand, the zinc particles were effectively melted, leading to better adhesion to the substrate. As previously reported, the contribution of mechanical and chemical aging to wash aging in the case of fabrics strongly depends on the substrate [32]. There are already several approaches to improve the durability of coatings. Sun et al. [33] described a technique to combine cotton or polymeric substrates with copper nanoparticles by polymer brushes. This extended the durability of the antibacterial coating. Another study by Ye et al. [34] showed excellent antibacterial durability without any chemical binders using chitosan-containing core–shell particles. The use of core–shell particles also proved to increase durability in our study. According to mechanical exposure through the crockmeter test, CS specimens showed the lowest loss of particle loading (Figure 8). Even after the washability treatment, the combination of copper and zinc had an advantage over the pure copper coating. Regardless of the performed treatment, the CS samples had a better lasting effect against *S. aureus* and ϕ6 (Figure 18). However, the best result after the washability treatment was achieved with the Zn1 coating. High exposure to a diluted acetic acid cleaning agent or 10,000 cycles did not lead to any change in the efficiency against *S. aureus*. Furthermore, an improved antiviral effect was even observed compared to the untreated surface (Figure 17). It is already known that Zn can have a better effect against Gram-positive and Gram-negative bacteria than Cu [35]. Regarding antiviral activity, the results appear to be vice versa. Our results of the untreated surfaces (Figure 14) as well as previous studies [36,37,38] showed that Cu has a better efficacy against viruses/bacteriophages than Zn. Moreover, many authors reported that the bactericidal activity of CuO or ZnO nanoparticles depends on their particle size and concentration. The authors of various studies concluded that a reduction in particle size increases the activity and enhances the powder concentration [35,39,40]. In our study, the Cu particles with an average diameter of 25 µm had the smallest size and the best effect without a surface treatment (Figure 14). The improved effect of Zn1 after the washability treatments can be explained by a better distribution of the particles on the surface (Figure 10) and the formation of larger clusters (Figure 12). A similar effect could be seen microscopically with the core–shell sample CS4. The crockmeter test led to an increase in particle loading of almost two percent (Figure 8), which was caused by flattening the particles through the application of the crockmeter. A slower viral effect was observed for the same CS specimen, which is why a washability test was not performed with this coating. An improved antimicrobial effect after the washability treatment as shown for Zn1 could therefore not be determined. However, the further aim of this study to determine the optimum CS particle loading was fulfilled. From a cost–benefit point of view, a particle loading of approximately 10% on PET-Al as in the case of CS3 proved to be the best variant in our setup. However, it should be noted that the test setup of the ISOs used for the bacteria or bacteriophage assay is very specific. For instance, the standard procedure for the incubation time points is carried out at very high humidity levels (~96% RH), which is often not adequate for the real environmental conditions of the final product. In addition, humidity can have an increasing or reducing effect on the antimicrobial efficacy [41]. However, there are not many alternative testing options. Maitz et al. recently compared a newly published ISO with the standard ISO 22196:2011. The authors described the new ISO 7581:2023 as not optimal and a test setup according to the end-use environments of the coated surfaces should be preferred over standard conditions [42]. Nevertheless, the used ISOs in our study are a good indication of the antimicrobial activity of non-porous surfaces.

## 5. Conclusions

To conclude, the CS particles with the combination of Zn core and Cu shell demonstrated several benefits in our study compared to the single-metal coatings. Better resistance against mechanical and chemical exposure, as well as adequate antimicrobial efficacy, especially after the washability treatment, could be assigned to the CS coating.

## Figures and Tables

**Figure 1 polymers-16-02209-f001:**
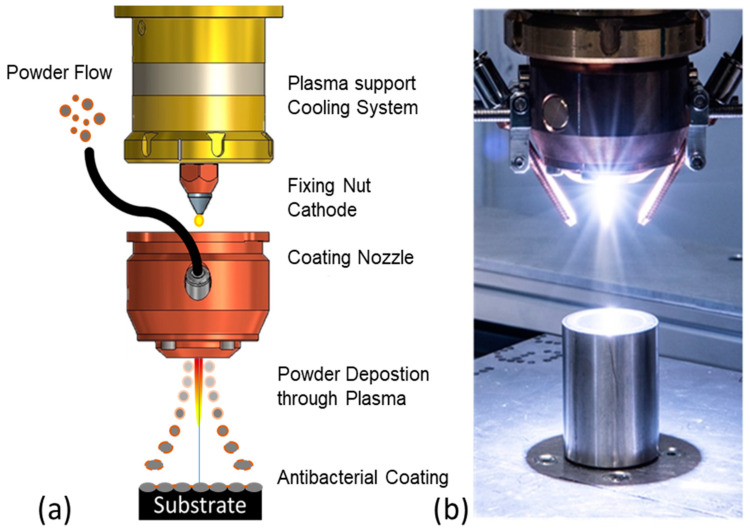
Schematic of the structure of the InoCoat 3 plasma nozzle and schematic coating process with powder (**a**); image of an activated plasma (**b**).

**Figure 2 polymers-16-02209-f002:**
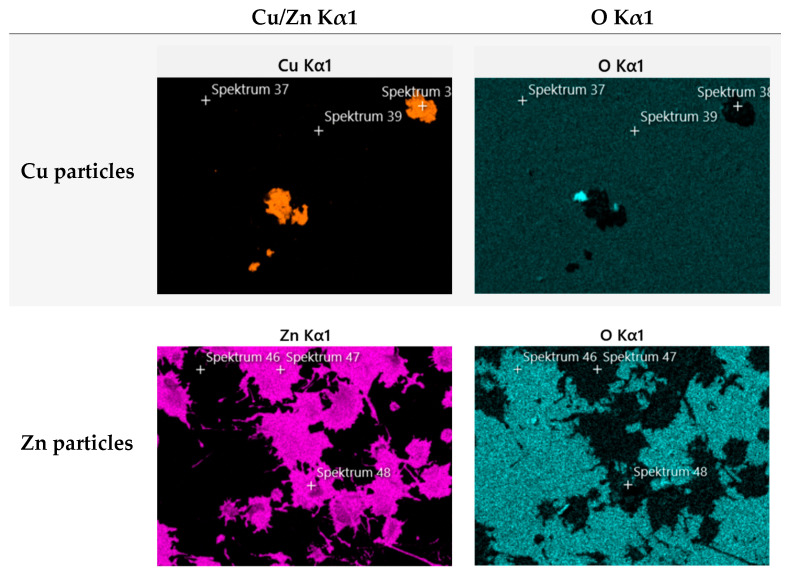
SEM/EDX images of the Cu and Zn layers: the oxygen and copper/zinc distribution can be recognized as complementary.

**Figure 3 polymers-16-02209-f003:**
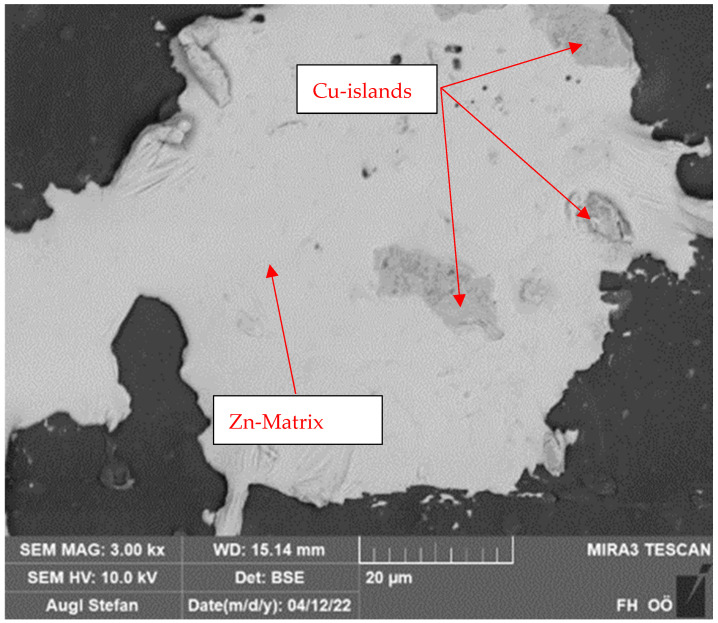
SEM close-up of the core–shell powder system.

**Figure 4 polymers-16-02209-f004:**
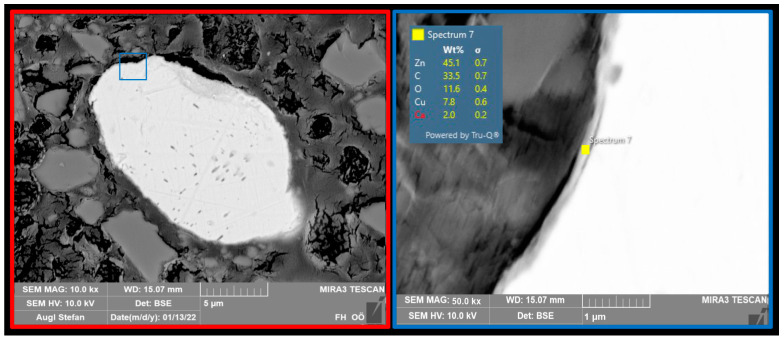
Cross-section of a CS particle and chemical composition of the shell.

**Figure 7 polymers-16-02209-f007:**
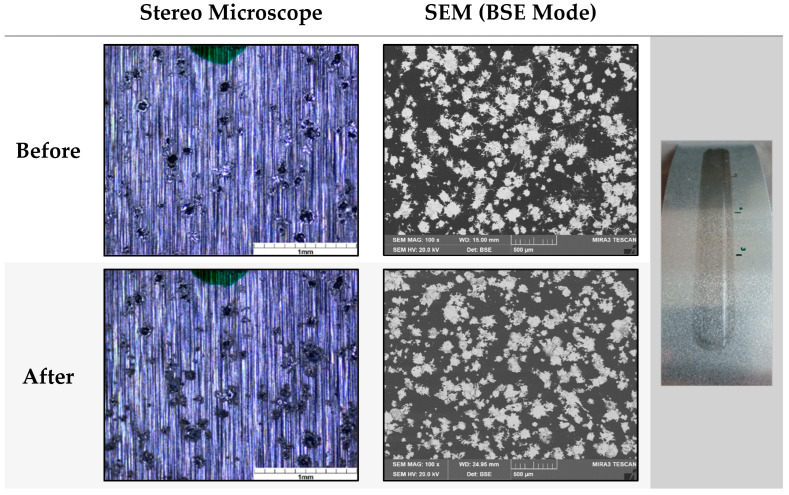
The analyses of the mechanical exposure from CS1 by comparing depictions before and after the treatment with the crockmeter. The differences between the normally used stereo microscopes (**left**) and SEM (**right**).

**Figure 8 polymers-16-02209-f008:**
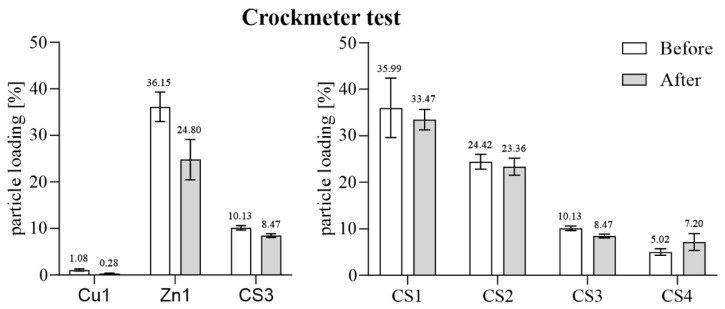
The results of the mechanical durability analyses by crockmeter test for the three types of coatings (**left**) and all CS coatings (**right**). After a 16 mm rubbing head ran over the surfaces for 1000 cycles with 9 N, the abrasion was analyzed optically in SEM with 7.5× magnification. The particle loading is indicated in percentage before and after the mechanical exposure.

**Figure 9 polymers-16-02209-f009:**
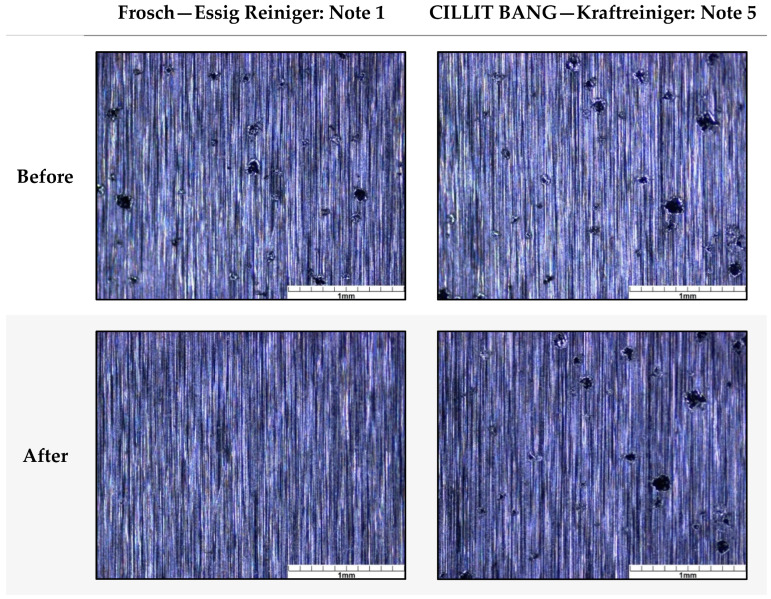
Comparison of the durability against two chemical cleaning agents with high (Frosch, note 1) and low (CILLIT BANG, note 5) impact on the surface of CS4. The analyses were performed with a stereo microscope.

**Figure 10 polymers-16-02209-f010:**
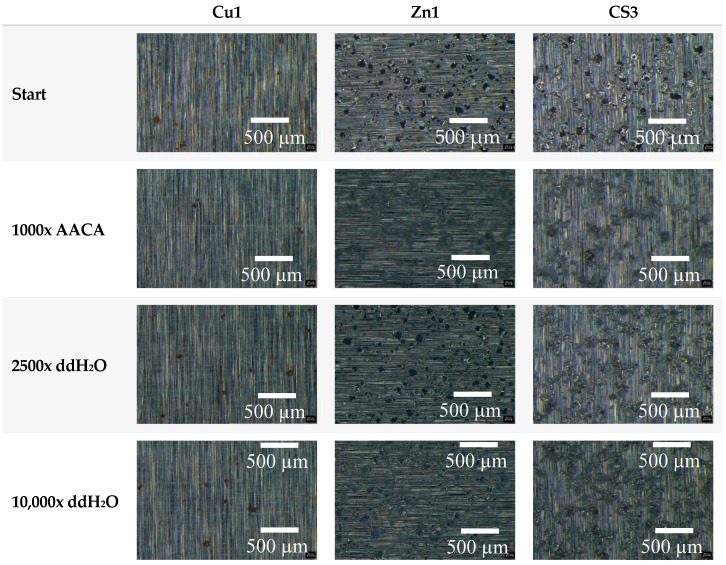
The results of the microscopic analyses through mechanical and chemical exposure within the washability test. The specimens Cu1, Zn1, and CS3 are shown at the start (untreated) as well as after the treatments with an acetic acid cleaning agent (AACA, 1000 cycles) and ddH_2_O (2500 cycles or 10,000 cycles). The samples were analyzed with 4× magnification in a stereo microscope.

**Figure 11 polymers-16-02209-f011:**
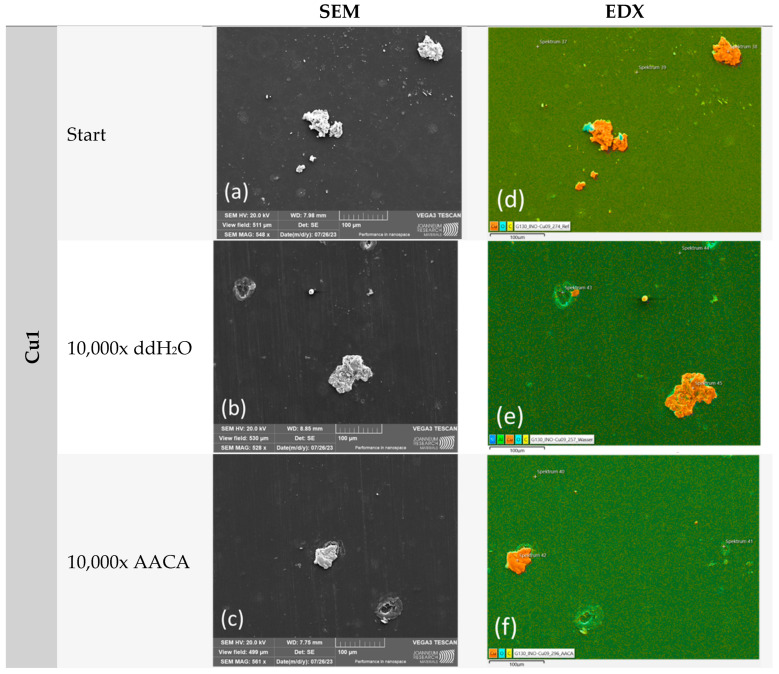
The washability test results of sample Cu1: (**a**,**d**) SEM and EDX results of the untreated sample (start); (**b**,**e**) after 10,000 cycles with ddH_2_O; and (**c**,**f**) after 10,000 cycles with AACA. The coating remained stable after both test methods.

**Figure 12 polymers-16-02209-f012:**
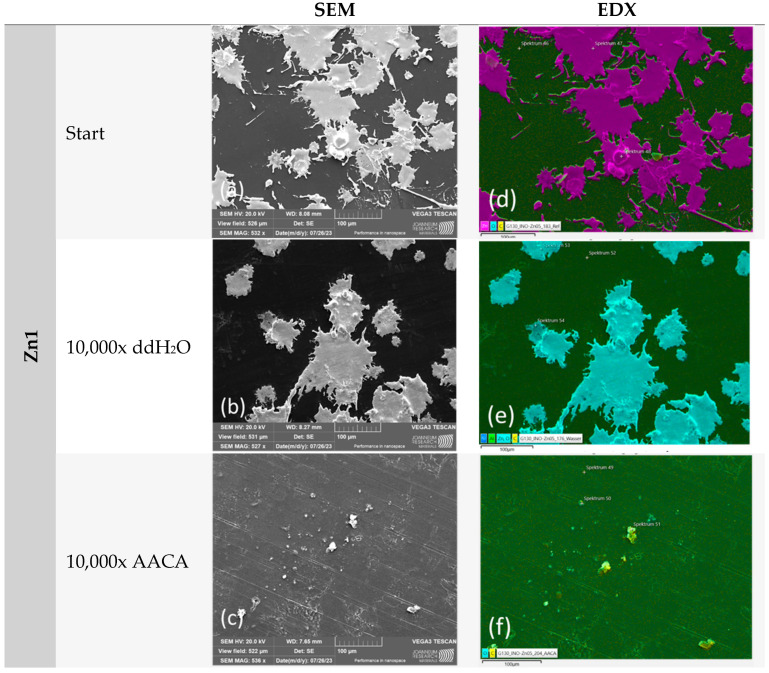
The washability test results of sample Zn1: (**a**,**d**) SEM and EDX results of the untreated sample (start); (**b**,**e**) after 10,000 cycles with ddH_2_O; and (**c**,**f**) after 10,000 cycles with AACA. The coating was only stable in the test method with ddH_2_O. After treatment with AACA, no more particles could be detected.

**Figure 13 polymers-16-02209-f013:**
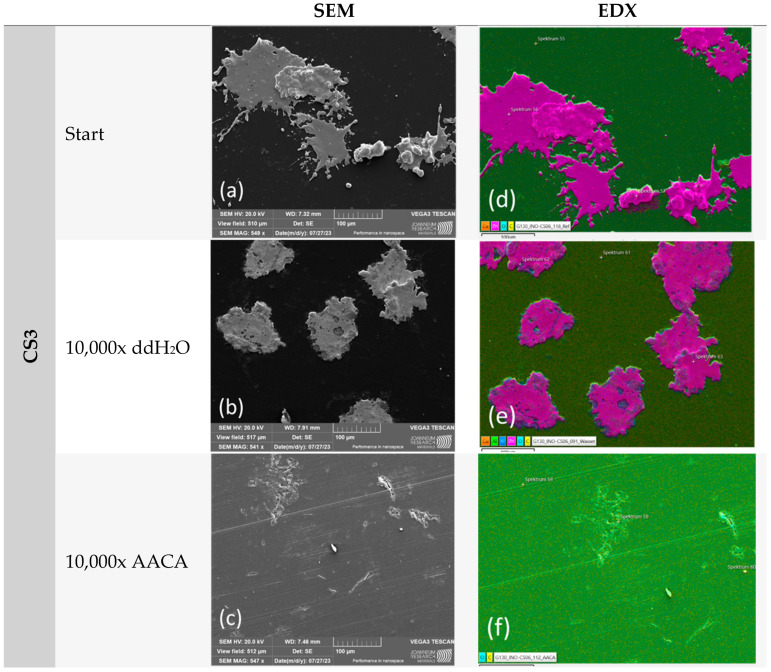
The washability test results of sample CS3: (**a**,**d**) SEM and EDX results of the untreated sample (start); (**b**,**e**) after 10,000 cycles with ddH_2_O; and (**c**,**f**) after 10,000 cycles with AACA. The coating was only stable in the test method with ddH_2_O. After treatment with AACA, no more particles could be detected.

**Figure 14 polymers-16-02209-f014:**
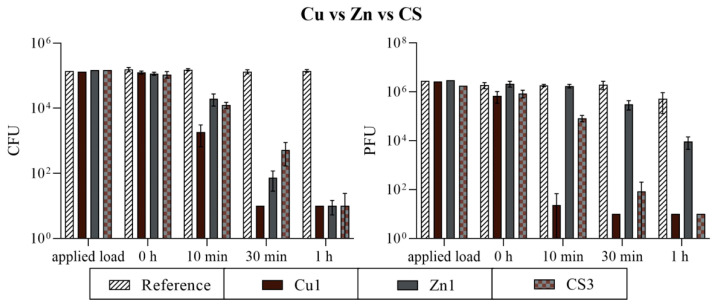
The quantification of *S. aureus* (**left**) and bacteriophage ϕ6 (**right**) after incubation on three different coating types. The surfaces coated with Cu, Zn, or CS were incubated with a bacterial load of approximately 10^5^ CFU/mL or a viral load of approximately 10^7^ PFU/mL (=applied load). The different temporal incubations (10 min, 30 min, and 1 h) were performed at 37 °C and a RH of ~96%. Further, 0 h shows the recovery immediately after inoculation on the tested samples. After the time points, the microorganisms were harvested and checked for survival. Uncoated PET-Al foils served as a reference. The error bars indicate the standard errors of the respective means, composed of triplicates (n = 3). The limit of detection was set as 10 CFU/PFU.

**Figure 15 polymers-16-02209-f015:**
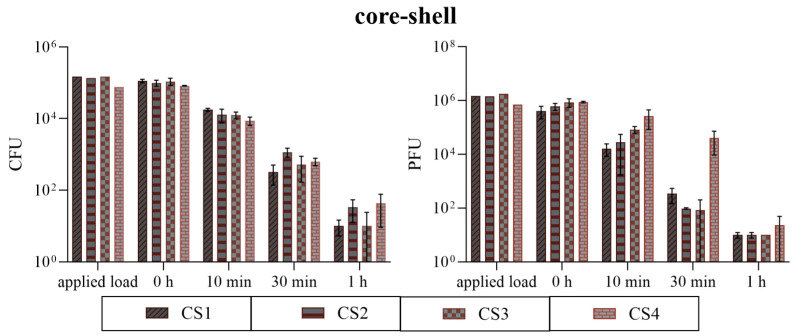
The quantification of *S. aureus* (**left**) and bacteriophage ϕ6 (**right**) after incubation on all core–shell coatings. The surfaces coated with different loading percentages of CS particles were incubated with a bacterial load of approximately 10^5^ CFU/mL or a viral load of approximately 10^7^ PFU/mL (=applied load). The different temporal incubations (10 min, 30 min, and 1 h) were performed at 37 °C and a RH of ~96%. Further, 0 h shows the recovery immediately after inoculation on the tested samples. After the time points, the microorganisms were harvested and checked for survival. The error bars indicate the standard errors of the respective means, composed of triplicates (n = 3). The limit of detection was set as 10 CFU/PFU.

**Figure 16 polymers-16-02209-f016:**
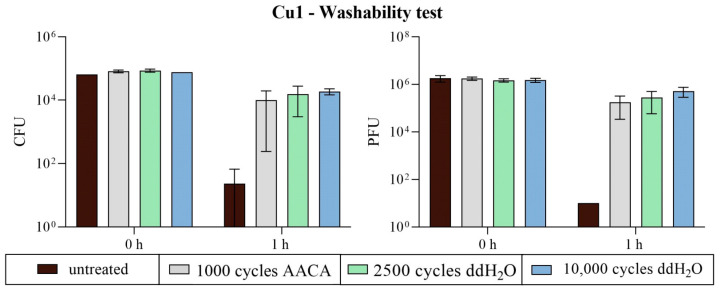
The antimicrobial efficiency of Cu1 after the washability test. The samples were compared before (untreated) and after the mechanical and chemical exposure tests using the Scrub Abrasion and Washability Tester AB6000. Different numbers of cycles and the addition of a diluted acetic acid cleaning agent (AACA with 1000 cycles) or ddH_2_O (2500 or 10,000 cycles) were selected as parameters. The results for the bacterial assay against *S. aureus* are given in CFUs (left), while PFUs (right) indicate the results for bacteriophage ϕ6. Two independent runs were carried out for the bacteria and bacteriophage assays, respectively. The average applied load was 8.1 × 10^4^ CFU and 2.7 × 10^6^ PFU. The one-hour incubation at 37 °C and an RH of ~96% was tested in triplicates in each case (n = 6). Each sample type’s initial concentration (0 h) was only determined twice. The averaged values with the corresponding standard deviation (if available) are indicated. The limit of detection was set as 10 CFU/PFU.

**Figure 17 polymers-16-02209-f017:**
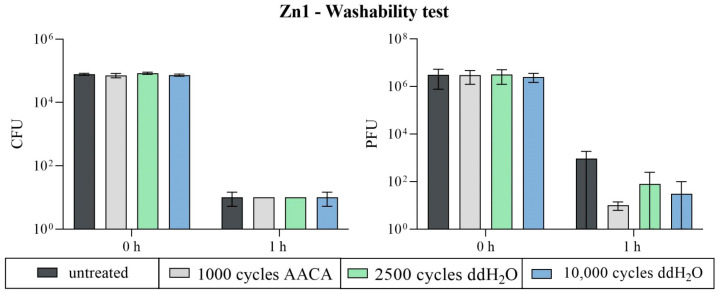
The antimicrobial efficiency of Zn1 after the washability test. The samples were compared before (untreated) and after the mechanical and chemical exposure tests using the Scrub Abrasion and Washability Tester AB6000. Different numbers of cycles and the addition of a diluted acetic acid cleaning agent (AACA with 1000 cycles) or ddH_2_O (2500 or 10,000 cycles) were selected as parameters. The results for the bacterial assay against *S. aureus* are given in CFUs (**left**), while PFUs (**right**) indicate the results for bacteriophage ϕ6. Two independent runs were carried out for the bacteria and bacteriophage assays, respectively. The average applied load was 6.2 × 10^4^ CFU and 3.3 × 10^6^ PFU. The one-hour incubation at 37 °C and an RH of ~96% was tested in triplicates in each case (n = 6). Each sample type’s initial concentration (0 h) was only determined twice. The averaged values with the corresponding standard deviation (if available) are indicated. The limit of detection was set as 10 CFU/PFU.

**Figure 18 polymers-16-02209-f018:**
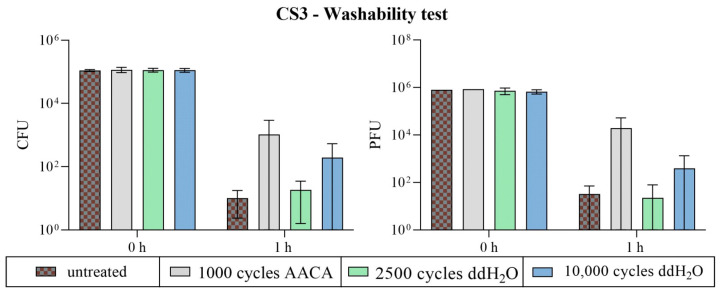
The antimicrobial efficiency of CS3 after the washability test. The samples were compared before (untreated) and after the mechanical and chemical exposure tests using the Scrub Abrasion and Washability Tester AB6000. Different numbers of cycles and the addition of a diluted acetic acid cleaning agent (AACA with 1000 cycles) or ddH_2_O (2500 or 10,000 cycles) were selected as parameters. The results for the bacterial assay against *S. aureus* are given in CFUs (**left**), while PFUs (**right**) indicate the results for bacteriophage ϕ6. Two independent runs were carried out for the bacteria and bacteriophage assays, respectively. The average applied load was 8.9 × 10^4^ CFU and 4.8 × 10^5^ PFU. The one-hour incubation at 37 °C and an RH of ~96% was tested in triplicates in each case (n = 6). Each sample type’s initial concentration (0 h) was only determined twice. The averaged values with the corresponding standard deviation (if available) are indicated. The limit of detection was set as 10 CFU/PFU.

**Table 1 polymers-16-02209-t001:** The samples tested in this study, with name, coating type, and particle loading percentage listed.

	Name	Coating	Particle Loading [%]
**PET-Al**	Reference	-	-
	Cu1	Copper particles	1.08
	Zn1	Zinc particles	36.15
	CS1	Core–shell particles	35.99
	CS2		24.42
	CS3		10.13
	CS4		5.02

**Table 2 polymers-16-02209-t002:** Household cleaners for chemical exposure.

	Frosch Essig Reiniger	Cif Bad & Dusche Salle de Bain	Frosch Glas-Reiniger Spiritus	Frosch Aktiv-Soda Reiniger	CILLIT BANG FETTLÖSER Kraftreiniger
**Application**	Removal of lime scale, water stains, soap residue, and dirt in all areas of the household	Cleaning of bathroom and shower	Cleaning of glass and smooth surfaces	Cleaning of almost all wipeable surfaces, cleaning of kitchen	Cleaning of all surfaces, cleaning of kitchen and bathroom
**pH**	2.4	3.9	5.5	10.4	11.4

**Table 3 polymers-16-02209-t003:** Evaluation criteria of the samples after the chemical exposure test.

Numerical Classification	Description
**5**	No change The test area cannot be distinguished from the adjacent area
**4**	Slight change The test area can only be distinguished from the adjacent area if the light source is reflected on the test surface and is reflected to the eye of the observer, e.g., a change in color or gloss No change in the surface structure, e.g., crack and blister formation
**3**	Moderate change The test area can be distinguished from the adjacent area, visible from several perspectives, e.g., a change in color or gloss No change in the surface structure, e.g., crack and blister formation
**2**	Substantial change The test area is clearly distinguishable from the adjacent area, visible from all perspectives, e.g., change in color or gloss and/or the surface structure has changed slightly, e.g., crack and blister formation
**1**	Major change The surface has changed considerably and/or change in color or gloss and/or the surface material has become partially or completely detached and/or filter paper adheres to the surface

**Table 4 polymers-16-02209-t004:** Results of chemical exposure to common household cleaning agents.

	Frosch	Cif	Frosch	Frosch	CILLIT BANG
	Essig Reiniger	Bad & Dusche Salle de Bain	Glas-Reiniger Spiritus	Aktiv-Soda Reiniger	FETTLÖSER Kraftreiniger
**CS1**	1.8	3.0	3.0	5.0	5.0
**CS2**	2.5	3.7	3.0	4.3	5.0
**CS3**	1.3	2.7	2.8	4.8	5.0
**CS4**	1.0	3.0	4.3	5.0	5.0

## Data Availability

The original contributions presented in the study are included in the article, further inquiries can be directed to the corresponding author.

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
