# Peer review of "Benefits of Core–Shell Particles over Single-Metal Coatings: Mechanical and Chemical Exposure and Antimicrobial Efficacy"

_polymers, 2024, doi:10.3390/polym16152209_

Round 1

Reviewer 1 Report

Comments and Suggestions for Authors

In this manuscript, the authors have explained nicely about the extensive usage of core-shell nanoparticles in medical applications.

Using crackmeter, the CS sample shows better resistance

The authors in the context has explained the chemical exposure for the sample. what is the optimum time exposed for all the 4 samples

Acetic acid acts as a cleaning agent here, since it is a weak acid. Do the authors can explain what is the changes observed when viewed in a strong acid.

Authors also explained about coating. What is the inference regarding the properties in the different coating environment. Authors can also add up regarding the coating thickness and it's influence.

Anti-moicrobial efficiency has well explained in the manuscript with gram positive and gram negative species

how do the Washability improves the surface property 

findings of particle size and concentration is very interesting to read

In APPD, the pressure optimization can be well explained

Anti-viral study for non porous surface is so informative

Author Response

Thank you for the response and the nice words to our study.

Comment 1: The authors in the context has explained the chemical exposure for the sample. what is the optimum time exposed for all the 4 samples

Response 1: This could be clarified by reducing time interval and looking on kill kinetics, which we did not do in this study.

Comment 2: Acetic acid acts as a cleaning agent here, since it is a weak acid. Do the authors can explain what is the changes observed when viewed in a strong acid.

Response 2: We used a compound system of polyethylene terephthalate and aluminium as substrate for the different coatings, which might not be very resistant against strong acids. Moreover, the CS coatings changed considerably by the vinegar cleaner. A stronger acidic agent would destroy the coating and the foil.

Comment 3: Authors also explained about coating. What is the inference regarding the properties in the different coating environment. Authors can also add up regarding the coating thickness and it's influence.

Response 3: Thickness was only determined for some powders we used. For CS particles it was about 100 nm thick (information added in the manuscript line 468). We did not focus on the thickness, but only on the load of the metal ions.

Comment 4: how do the Washability improves the surface property

Response 4: The washability treatment flattens the particles of Zn and leads to surfaces enlargement, which definitively increases the chance of contact of the microorganisms to the active ingredient (see line 314-319)

Reviewer 2 Report

Comments and Suggestions for Authors

Comments to Authors

In this manuscript, authors report the core-shell particles assembly over single-metal coatings assessed by mechanical & chemical exposure and antimicrobial activities. Polyethylene terephthalate and aluminum (PET-Al) foils were coated with copper, zinc and a combination of these two metals, which was referred as core-shell particles. The best efficacy was achieved with the pure copper coating, which was not convincing in the abrasion tests. In addition, the antimicrobial efficiency remained more stable after the washability treatment.

The manuscript does not meet the merits of Polymers owing to the following reasons:

1.      The project has not been well planned. Even the necessary measurements have not been performed.

2.      The overall research contents of the manuscript are highly inadequate and did not offer valuable research progress.

3.      Most of the findings/conclusions/statements are not supported by provided data.

4.      There are also various mis-leading and confusing terminologies (like core-shell) in the manuscript which even have no evidences of their existence in this work.

5.      The provided results are ordinary and very limited, with no strong evidences and worthful findings.

Comments on the Quality of English Language

N/A

Author Response

Thank you for the criticism on our manuscript but the comments are not helpful and did not give us information that could lead to an improvement of our manuscript.

Reviewer 3 Report

Comments and Suggestions for Authors

Public surfaces serve as a permanent reservoir for infectious microorganisms, which is a growing problem in areas of daily life. Coating surfaces with inorganic antimicrobials such as copper, zinc, or combinations thereof, can help reduce adhesion and microbial growth. In this study, polyethylene terephthalate and aluminum foils were coated with copper and zinc, and a combination of these two metals known as core-shell particles, respectively, were used. The aim was to study the antibacterial characteristics of the system after atmospheric plasma spraying. For this purpose, metal powders were deposited using an atmospheric plasma spray system that was preheated to various temperatures. Various application characteristics were investigated, such as antibacterial properties, adhesion strength, and coating hardness. All deposition parameters were fixed, except the substrate temperature.

Previously, it was reported on the use of an air plasma jet with direct current to apply thin coatings of copper and zinc to polymers. However, the novelty of this work lies in the comparison of the zinc/copper binary system to a combination of zinc core and copper shell under conditions of abrasion and water treatment. Unfortunately, the authors did not provide detailed information on the core-shell particles, making it difficult to assess the contribution of each metal individually. Questions arise regarding the effectiveness of a zinc-core coated with copper, or the degree to which the copper coats the zinc core.

Comparative studies of microbial contamination under conditions simulating wear, abrasion or water treatment of the surface have shown that core-shell particles coating have a significant antimicrobial effect compared to zinc and are slightly inferior to copper. The results presented allow us to consider the developed core-shell coating for polymers as an effective method for combating health-related infections. Nevertheless, the authors avoided the question of environmental friendliness of this coating. What happens to the heavy metal particles after they leave the surface? There was no chemical analysis of the washing fluid or estimation of metal cation concentration on the carrier surface.

The described materials have shown their potential use as a biocidal coating, which can be of great importance for reducing microbial load on public surfaces, such as in medical institutions, to prevent infection transmission and protect materials from degradation. In addition, it is worth noting the advantage of the compatibility with various materials, including textiles, which provides improved core-shell properties for particle applications.

Comments:

1. It is necessary to describe plasma source and plasma treatment in more detail.

2. Under atmospheric plasma conditions, a mixture of metal particles and metal oxides forms. Data on the chemical composition of metal compounds on the surface is needed, since antimicrobial activity of metals and metal oxide differs significantly. In addition, data on the dissolution of particles in acidic medium is more likely to correspond to oxide, since a protective film forms on metal particle surface after plasma application.

3. It is necessary to evaluate the ratio of zinc and copper surfaces to the core and shell of the particle. Antimicrobial activity depends on combination of Zn core and Cu shell. When fully coated, role of zinc is negligible. In "Methods" section add description of method for determining available surface of Cu and free surface of Zn in CS particles.

4. For a more accurate assessment of particle size in the manuscript, histograms need to be shown. For ease of comparison, measured height distributions are better represented in the form of cumulative distributions, which are integrals of distribution densities.

5. Information on the presence of particles or ions of heavy metals in washing liquids is needed to prove the environmental friendliness of developed antimicrobial 

Author Response

Thank you for your comments on our manuscript.

Comment 1: It is necessary to describe plasma source and plasma treatment in more detail.

Answer 1: We included the requested information (Line 435-444)

Comment 2: Under atmospheric plasma conditions, a mixture of metal particles and metal oxides forms. Data on the chemical composition of metal compounds on the surface is needed, since antimicrobial activity of metals and metal oxide differs significantly. In addition, data on the dissolution of particles in acidic medium is more likely to correspond to oxide, since a protective film forms on metal particle surface after plasma application.

Answer 2: Data on the chemical composition was added (Line 449-458 for Cu and Zn coating, Line 459-478 for CS coating); Oxidation processes can of course be triggered by the acidic conditions with the vinegar cleaner used and thus additionally accelerate the layer detachment, but during the project we didn’t observe that.

Comment 3: It is necessary to evaluate the ratio of zinc and copper surfaces to the core and shell of the particle. Antimicrobial activity depends on combination of Zn core and Cu shell. When fully coated, role of zinc is negligible. In "Methods" section add description of method for determining available surface of Cu and free surface of Zn in CS particles.

Answer 3: Ratio of Cu and Zn and uncoated areas for CS has been added (Line: 459-466)

Comment 4: For a more accurate assessment of particle size in the manuscript, histograms need to be shown. For ease of comparison, measured height distributions are better represented in the form of cumulative distributions, which are integrals of distribution densities.

Answer 4: The particle size is already described in the material and methods section (Line: 425-429). Unfortunately, further information couldn’t  be provided by the industrial partner due to competitive situation.  

Comment 5:  Information on the presence of particles or ions of heavy metals in washing liquids is needed to prove the environmental friendliness of developed antimicrobial

Answer 5: These analyses are very interesting, but were not carried out due to the lack of necessary equipment. The washability tests were used to check the stability of the coatings and to determine the influence of the "cleaning agents" used.

Round 2

Reviewer 3 Report

Comments and Suggestions for Authors

I`m grateful to the authors for their work to eliminate my comments.